# Development of support material for health professionals who are implementing Shared Decision-making in breast cancer screening: validation using the Delphi technique

María José Hernández-Leal ,[1,2,3] Núria Codern-Bové,[4] María José Pérez-Lacasta,[1,3] Angels Cardona,[5] Carmen Vidal-Lancis,[6] Misericòrdia Carles-Lavila,[1,2,3] on behalf of the ProShare Group

For numbered affiliations see end of article.

**Correspondence to**
Professor Misericòrdia Carles-Lavila;
misericordia.carles@urv.cat

## ABSTRACT

**Background** The Literature is no report support material on Shared Decision-making applied to breast cancer screening that is intended for Spanish health professionals. The researcher created both a handbook and a guide for this topic using an adaption of the Three-talk model.

**Objective** A Delphi method will be used to reach an agreement among experts on the contents and design of a manual and guide, designed by the research team, and to be used by health professionals in the application of SDM in breast cancer screening.

**Design** A qualitative study. The content and design of the handbook and the guide was discussed by 20 experts. The Delphi techniques was in an online mode between July and October 2020 and researchers used Google forms in three rounds with open and closed questions. The criterion established for consensus was a coefficient of concordance (Cc) above 75, for questions using a Likert scale of 1–6—in which 1 meant 'completely disagree' and 6 'completely agree'—with a cut-off point equal to or higher than 4.

**Results** Participants considered the Three-talk model suitable for the screening context. The handbook sections and level of detail were considered satisfactory (Cc=90). The summary provided by the clinical practice guide was considered necessary (Cc=75), as it was the self-assessment tool for professionals (Cc=85). Content was added: addressing the limitations of the SDM model; extending the number of sample dialogues for health professionals; providing supplementary resources on using Patient Decisions aids and adding references on communication skills.

**Conclusions and applications** The first handbook and clinical practice guide providing unique SDM support material for health professionals have been developed. The handbook and guide are useful and innovative as supporting material for health professionals, but training strategies for SDM and a piloting plan for the use of materials are requested, in order to facilitate its implementation.

## Strengths and limitations of this study

► Development of a handbook and a clinical practice guide on Shared Decision-making for breast cancer screening.
► Adaptation of the Three-talk model for breast cancer screening.
► Participation of professionals in validating the design of the support materials.
► Facilitating the application of a person-centred model to the screening context.

## BACKGROUND

Shared Decision-making (SDM) is recommended in an uncertainty context—among others—in which it is necessary to argue on the risks and benefits, in the health topics.[1] SDM is a doctor-patient relationship model, and both collaborate to deliberate over the best choice based not only on scientific evidence but also on women's preferences and values.[2 3] Thus, SDM invites you to change the paternalistic health model for a more participatory one, seeking patients' greater involvement in their health, instead of aiming at a greater adherence to treatments, procedures or medicines, even though it has also been associated as a result of its application.[4]

In Spain, Law 21/2000 on health information rights, patient autonomy and clinical documentation[5] protects the right to decide freely. However, SDM is not explicitly recommended for screening programmes. And the scientific community is making efforts to create patient decisions aids (PtDAs)[6 7] to be integrated in the early detection programmes of autonomous communities, but, at the moment, its use is not widespread.

The breast cancer screening programme currently falls under the oncology master plan (*Plan Director de Oncologí*(a) in Catalonia.[8] However, while there are strategies for incorporating women's values and preferences into the decision on whether to have the examination or not, there is no associated framework on how to put them into practice.[9] The current situation in Catalonia is this: the breast cancer detection programme (*Programa de Detección del Cáncer de Mam*a) sends—every 2 years— women between 50 years and 69 years of age a letter informing them of the time and date when they should attend their local health centre to have a mammogram.[10] The programme achieves a high level of coverage, but it fails to incorporate an opportunity for women and professionals to exchange information and have a dialogue on her decision. To promote women's participation, several research teams have developed projects that involve women in making their decision on screening. In 2017, Toledo-Chavari and their colleagues created PtDAs[5] (online supplemental file 1), consisting of a trifold leaflet that provided balanced information on either the benefits or adverse effects, for both professionals and women, to be used during the clinical appointment. However, based on the barriers and enabling factors cited in the literature,[11–13] the researchers decided not to use the PtDAs alone, for it was not enough, and concluded that SDM training material aimed at health professionals was also needed. The manual is training material, since it is a useful tool to transmit knowledge and provide quick and simple information on how to operationalise new practices, introducing beginners into the theme on how to use it the same way advanced users do.[14] Considering that SDM is not a common practice, a manual could, to some extent, fill knowledge gaps on this model.

The ProShare Study. Our research team has therefore developed a handbook—manual—(online supplemental file 2) and guide[15] (online supplemental file 3) aimed at health professionals who have a direct relationship with women. These documents should be used as reference material by health professionals when discussing the decision with women on whether to perform—or not to—a mammography, taking into consideration key elements and providing the patient with: information and education, and interpersonal communication between doctor and patient for a final decision.[16] To facilitate the implementation of SDM, the model used as a reference was the Three-Talk model. The model was created so that three key steps (1—Team Talk, 2—Option Talk, 3—Decision Talk) would be quickly grasped and to explain in an easy way how to apply SDM in a generic health context for healthcare professionals.[17] In this article we are adapting the three steps of the model to a specific health context in breast cancer screening to: (1) Team talk; (2) Option talk and exploring preferences; (3) Decision talk. A self-assessment of SDM was included in the manual, which should be applied at the end of the appointment so that professionals can identify strengths and weaknesses in the implementation of the SDM. Finally, the guide provides a summary of the handbook to be used in the same appointment as a reminder of the three steps.

The objective of this study is to use a Delphi method to reach an agreement among experts on the contents and design of a manual and guide, designed by the research team and to be used by health professionals in the application of SDM in breast cancer screening.

## METHODS
### Delphi technique
The main objective of the Delphi technique is to arrive at a consensus among experts on specific topics. For this reason, the researchers decided to use this as it was necessary to consult highly qualified people to develop a supporting material for a topic little explored and therefore insufficient research. Thus, seeking the opinion of experts is a common approach[18] and in this case experts are required for the development of a manual and guide because there are few documents focused on health professionals explaining the application of SDM, specifically for breast cancer screening. Another feature of the Delphi technique is that participants undergo a series of online survey question rounds, which are formulated with elements not agreed on in the previous round.[19 20] This process is repeated continuously until one of the completion criteria is met.[21] A further requirement for the formulation of the Delphi technique is that the responses of all experts must be shared in each round, allowing experts to reassess their responses in the light of other experts' views. Finally, all the rounds must be carried out anonymously to ensure that they do not influence others just because of one expert's considerable knowledge on the topic. One of the limitations the Delphi technique has is that it provides experts' opinion; however, other complementary techniques could also be considered to determine a final position on the subject of the study.[18–20] The experts participating in a virtual way can overcome barriers related to economic circumstances and geographical or time-related constraints.[19 20] Experts, according to literature, can be grouped into two broad categories: *Subjects (Su)* —people who would use the instrument in their profession; and *Specialists (Sp)*—people who have knowledge about the subject due to their academic and/ or professional experience.[19 20]

### Participants
The handbook and clinical practice guide, entitled 'The participation of health professionals in Shared Decision-Making in breast cancer screening' (*La participación de los profesionales de la salud en la Toma de Decisiones Compartida en el cribado de cáncer de mam*(a) (online supplemental files 4 and 5),[15] were developed by the Pro-Share research team. The first version was produced with the participation of three researchers with experience in SDM and breast cancer screening, who acted as external reviewers, and two health professionals, who designed the plan for piloting the questionnaire online (Google form).

The included criteria for participants were as follow*s:*

► *Subjects*: (a) Health professionals, preferably from primary care services, who provide direct care to women through breast cancer prevention activities, and (b) Health professionals, who have at least 5 years' experience[21] in the Spanish Health System.

► *Specialists*: (a) International-level researchers whose research career has focused on the SDM model, and (b) Those who are proficient in Spanish (given that the handbook has been produced in Spanish). Preference was given to individuals who had developed educational support material for professionals.[21]

Literature was consulted to determine the sample size. It mentions that very large sizes (more than 50 participants) could prevent reaching agreements in limited time., I . Moreover, it depends on the heterogeneity of the experts. If they are from different countries or various specialisations, they enrich the opinions formulated.[20] Therefore, a limit between 7 and 30[20] was decided, most commonly being a total of 15 to 20 experts.[20]

### Patient and public involvement

Health professionals and researchers gave feedback on the Delphi rounds about the manual and guide practice, which were adapted accordingly.

### Procedure and data collection

The two sampling strategies of the researchers were used to recruit participants: convenience sampling for specialists and snowball sampling for health professionals. For specialists, the researchers were looking for published articles about SDM and contacted the authors via email (MJHL, MCL, MJPL). For health professionals, researchers sent an email with an invitation to NCB and AC, and they could be again sent to other colleagues. Finally, the researchers (NCB and AC) sent invitations via email to 43 potential experts to participate in a Delphi round; 30 of them accepted. The aim was to determine the usefulness of the topics, the relevance of the content and the designed document of the material for the SDM on breast cancer screening. The Delphi round was being done on Google forms between July and October 2020.

For round 1, open and close questions were considered with topics relevant to the research objective *'The sections of the handbook are effective for understanding the application of SDM to breast cancer screening'* or *'Do you think that a guide concisely summarising the SDM steps is necessary?'*. Participants had to mark the degree of agreement to the questions using a Likert Scale of 1–6, in which 1 was 'completely disagree' and 6 was 'completely agree'. Later, the researchers (MJHL, MCL, MJPL) sent a anonymous report to the experts with the responses of all of them, so that they would consider their opinion, especially on those questions that did not reach a level of agreement (coefficient of concordance (Cc)<75). Disagreement questions were raised again in the following rounds until the

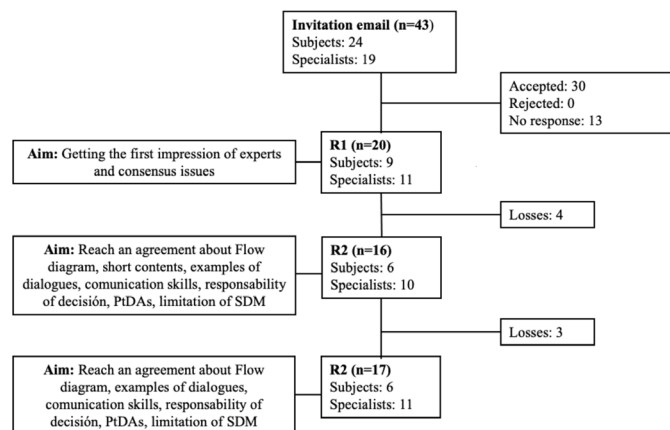

**Figure 1** Flow diagram of participation in each round. SDM, shared decision-making; PtDAs, Patient Decisions aids.

necessary agreement was reached in most transverse aspects. This was finally achieved in round 3.

### Data analysis

The researchers (MJHL, MCL, MJPL, NCB, AC) analysed participants' responses at the end of each round, considering the responses in which the score on the Likert Scale was 4 or above to be positive. Agreement was determined to be reached when Cc >75.[22] . For calculation consider the next formula:

$$Cc = \left(1 - \frac{Vn}{Vt}\right) x100,$$

Vn=number of negative votes (a score of lower than 4); Vt=total number of votes (n=6).[22]

The researchers considered the Martínez-Piñedo criteria for closing the Delphi in round 3.[23,23].

Informed consent was secured, which stated that participants accepted the conditions of participation on agreeing to respond to the questionnaire. These conditions specified that responses were confidential and would only be used for the purposes of this research.

### RESULTS

Out of the 30 professionals who initially agreed to participate, 20 (66.6%) went on to respond in the first round (R1), 16 (53.3%) in the second one (R2) and 17 (56.6%) in the third one (R3) (figure 1). In R1, the mean age of the experts was 46.6 years (SD 10.25), 75% were female, 65% were doctors, 70% worked in the public sector and they had on average 19 years' (SD 9.69) experience (table 1).

From the outcomes of the Delphi technique an agreement on the content and design of the documents could be reached. Among the three rounds carried out, four significant changes were made regarding the contents: (1) Including examples of practical dialogues for each phase, (2) Annexed additional information on communicative skills, (3) Incorporation of information on how to manage professionals' responsibility in SDM, (4) Additional information about limitations of the SDM

**Table 1** Characteristics of the participants

| Variable | | Round 1 | | Round 2 | | Round 3 | |
| --- | --- | --- | --- | --- | --- | --- | --- |
| | | N | % | N | % | N | % |
| Sex | Female | 15 | 75 | 12 | 75 | 13 | 76.47 |
| | Male | 5 | 25 | 4 | 25 | 4 | 23.52 |
| | Total | 20 | 100 | 16 | 100 | 17 | 100 |
| Age range (years) | 31–40 | 7 | 35 | 7 | 43.75 | 7 | 41.17 |
| | 41–50 | 6 | 30 | 4 | 25 | 5 | 29.41 |
| | 51–60 | 5 | 25 | 4 | 25 | 4 | 23.52 |
| | 61–70 | 2 | 10 | 1 | 6.25 | 1 | 5.88 |
| | Total | 20 | 100 | 16 | 100 | 17 | 100 |
| Ownership of the affiliated institute, health centre or research site | Public sector | 14 | 70 | 11 | 68.75 | 11 | 64.7 |
| | Private sector | 6 | 30 | 5 | 31.25 | 6 | 35.29 |
| | Total | 20 | 100 | 16 | 100 | 17 | 100 |
| Profession | Nursing | 4 | 20 | 2 | 12.5 | 3 | 17.64 |
| | Medicine | 13 | 65 | 11 | 68.75 | 11 | 64.7 |
| | Psychology | 1 | 5 | 1 | 6.25 | 1 | 5.88 |
| | Other | 2 | 10 | 2 | 12.5 | 2 | 11.76 |
| | Total | 20 | 100 | 16 | 100 | 17 | 100 |
| Specialty | Family and community medicine or nursing | 14 | 70 | 11 | 68.75 | 12 | 70.58 |
| | Public health | 1 | 5 | 1 | 6.25 | 1 | 5.88 |
| | Gynaecology | 1 | 5 | 1 | 6.25 | 1 | 5.88 |
| | Endocrinology | 1 | 5 | 1 | 6.25 | 1 | 5.88 |
| | Research in health services | 1 | 5 | 1 | 6.25 | 1 | 5.88 |
| | Content development for decision support systems for healthcare | 1 | 5 | 1 | 6.25 | 1 | 5.88 |
| | None | 1 | 5 | 0 | 0 | 0 | 0 |
| | Total | 20 | 100 | 16 | 100 | 17 | 100 |
| Experience (years) | 6–10 | 6 | 30 | 6 | 37.5 | 6 | 35.29 |
| | 11–20 | 6 | 30 | 5 | 31.25 | 6 | 35.29 |
| | 21–30 | 6 | 30 | 5 | 31.25 | 5 | 29.41 |
| | 31–40 | 2 | 10 | 0 | 0 | 0 | 0 |
| | Total | 20 | 100 | 16 | 100 | 17 | 100 |

model and (5) Elimination of the flow of the screening programme in Catalonia.

It was impossible to determine why professionals changed their decisions in the rounds, since they only had options to change their vote once the results of the previous rounds were known and their peers' arguments were read. The results for each round are as follows.

### Round 1

R1 was designed to achieve two objectives: determine its utility and clarify the content and the design of the supporting material. For this purpose, participants were asked 33 Likert scale questions, 1 multiple-choice question and 6 open questions on the handbook and they were also given 2 Likert Scale questions and 4 open questions on the clinical practical guide (table 2).

A Cc higher than 75 was recorded for 32 of the Likert Scale questions and the minimum Cc was not reached by only 3 of them; in other words, no agreement was reached. These questions concluded that 'Flow diagram of the Early Detection of Breast Cancer programme', was clear (Cc=60) and useful (Cc=70). The same applied to the question that determined Team talk (page 34)—to be clear (Cc=75). These questions were incorporated into R2.

In the multiple-choice question, participants were asked which section of the handbook should be edited: 10 responded 'none'; 5 chose the section entitled 'Which skills or competencies do health professionals need?'; 3 chose the 'Screening programme' section and 2 chose the 'Introduction' (figure 2).

In their open responses, most participants considered the initiative to be positive and thought it would enable health professionals to access information on SDM using the Three-talk model in breast cancer screening (box 1).

**Table 2** R1 responses

| Section | Questions using a Likert scale of 1 (completely disagree) to 6 (completely agree) | 1 | 2 | 3 | 4 | 5 | 6 | Cc |
|---|---|---|---|---|---|---|---|---|
| Evaluation of the handbook on Shared Decision-making in breast cancer screening | 1. The sections of the handbook are effective for understanding the application of SDM to breast cancer screening | 0 | 5 | 5 | 10 | 55 | 25 | 90 |
| | 4. The 'Contents' section is suitable for this handbook | 0 | 5 | 5 | 5 | 35 | 50 | 90 |
| | 5. The 'Objective of the material' section is suitable for this handbook | 0 | 5 | 0 | 5 | 30 | 60 | 95 |
| | 6. The 'Who is it aimed at?' section is suitable for this handbook | 0 | 5 | 0 | 10 | 35 | 50 | 95 |
| | 7. The 'Introduction' section is suitable for this handbook | 5 | 10 | 0 | 10 | 45 | 30 | 85 |
| | 8.The 'Shared Decision-Making: What is it?' section is suitable for this handbook | 0 | 5 | 5 | 10 | 40 | 40 | 90 |
| | 9.The 'Shared Decision-Making: Why is it important?' section is suitable for this handbook | 0 | 5 | 0 | 5 | 45 | 45 | 95 |
| | 10.The 'Shared Decision-Making: What skills or competencies do health professionals need?' section is suitable for this handbook | 0 | 5 | 5 | 35 | 35 | 20 | 90 |
| | 11.The 'Shared Decision-Making: What do patients think?' section is suitable for this handbook | 0 | 10 | 0 | 10 | 35 | 45 | 90 |
| | 12.The 'Shared Decision-Making in breast cancer screening: The screening programme' section is suitable for this handbook | 0 | 10 | 10 | 5 | 30 | 45 | 80 |
| | 13. The 'Shared Decision-Making in breast cancer screening: Implementation of SDM in breast cancer screening' section is suitable for this handbook | 0 | 5 | 0 | 5 | 45 | 45 | 95 |
| | 14. The 'Shared Decision-Making in breast cancer screening: Self-assessment of the SDM process' section is suitable for this handbook | 0 | 10 | 10 | 10 | 35 | 35 | 80 |
| | 15. The handbook provides the minimum content on SDM in breast cancer screening that health professionals should be familiar with | 0 | 5 | 0 | 20 | 50 | 25 | 95 |
| | 16. The content of the handbook is sufficiently detailed | 0 | 5 | 5 | 5 | 35 | 50 | 90 |
| | 19 .a. : Models of healthcare (page 14) is useful | 0 | 0 | 10 | 15 | 30 | 45 | 90 |
| | 19.b. : Models of healthcare (page 14) is clear | 0 | 0 | 15 | 10 | 20 | 55 | 85 |
| | 20.a. : Role of the participants in the clinical encounter (page 15) is useful | 0 | 5 | 5 | 10 | 35 | 45 | 90 |
| | 20.b. : Role of the participants in the clinical encounter (page 15) is clear | 0 | 5 | 5 | 15 | 30 | 45 | 90 |
| | 21.a. : Elements of SDM (page 16) is useful | 0 | 0 | 10 | 20 | 25 | 45 | 90 |
| | 21.b. : Elements of SDM (page 16) is clear | 0 | 0 | 5 | 20 | 25 | 50 | 95 |
| | 22 .a. : Communication skills (page 21) is useful | 5 | 0 | 10 | 30 | 15 | 40 | 85 |
| | 22.b. : Communication skills (page 21) is clear | 0 | 0 | 10 | 20 | 25 | 45 | 90 |
| | 23 .a. : Flow diagram of the early detection of breast cancer programme (page 27) is useful | 5 | 5 | 20 | 15 | 20 | 35 | 70 |
| | 23.b. : Flow diagram of the early detection of breast cancer programme (page 27) is clear | 10 | 10 | 20 | 15 | 15 | 30 | 60 |
| | 24 .a. Team talk (page 34) is useful | 10 | 5 | 0 | 30 | 25 | 30 | 85 |
| | 24.b. Team talk (page 34) is clear | 10 | 0 | 15 | 20 | 25 | 30 | 75 |
| | 26 .a. Option talk (page 36) is useful | 5 | 5 | 0 | 30 | 40 | 20 | 90 |
| | 26.b. Option talk (page 36) is clear | 5 | 0 | 10 | 45 | 15 | 25 | 85 |

**Table 2** Continued

| Section | Questions using a Likert scale of 1 (completely disagree) to 6 (completely agree) | 1 | 2 | 3 | 4 | 5 | 6 | Cc |
|---|---|---|---|---|---|---|---|---|
| | 28.a. Decision talk (page 38) is useful | 0 | 5 | 10 | 5 | 35 | 45 | 80 |
| | 28.b. Decision talk (page 38) is clear | 0 | 0 | 5 | 15 | 30 | 50 | 95 |
| | 30.a. SDM steps (page 39) is useful | 0 | 0 | 5 | 20 | 25 | 50 | 95 |
| | 30.b. SDM steps (page 39) is clear | 0 | 0 | 10 | 35 | 10 | 45 | 90 |
| | 31.Does its design (colours, images) make the handbook easier to read for an SDM professional? | 0 | 0 | 5 | 20 | 35 | 40 | 95 |

| Closed questions | Options | Percentage (%) |
|---|---|---|
| 2. Which section of the handbook do you think should be changed? | (a) Front cover | 0 |
| | (b) Objective of the material | 0 |
| | (c) Who is it aimed at? | 0 |
| | (d) Introduction | 10 |
| | (e) SDM: What is it? | 0 |
| | (f) SDM: Why is it important? | 0 |
| | (g) SDM: What skills or competencies do health professionals need? | 25 |
| | (h) SDM: What do patients think? | 0 |
| | (i) SDM in breast cancer screening: The screening programme | 15 |
| | (j) SDM in breast cancer screening: Implementation of SDM in breast cancer screening | 0 |
| | (k) SDM in breast cancer screening: Self-assessment of the SDM process | 0 |
| | (l) None | 50 |
| | Total | 100 |

| Section | Questions using a Likert scale of 1 (completely disagree) to 6 (completely agree) | 1 | 2 | 3 | 4 | 5 | 6 | Cc* |
|---|---|---|---|---|---|---|---|---|
| Evaluation of the Clinical Practice Guide: Implementation of SDM for Healthcare Professionals | 1. Do you think that a clinical practice guide concisely summarising the SDM steps is necessary? | 0 | 15 | 10 | 5 | 5 | 65 | 75 |
| | 6. Is it useful to incorporate the self-assessment section in the clinical practice guide? | 0 | 5 | 10 | 25 | 25 | 35 | 85 |

Cc, coefficient of concordance; SDM, shared decision-making.

**Figure 2** Changes made to the index.

However, one of the participants suggested using the Agency for Healthcare Research and Quality model.

The participants also provided some suggestions to modify the handbook. The most frequently cited were concerned with the length of the handbook and recommended simplifying the content (box 2) and incorporating sample dialogues, communication skills (box 3) and instructions for using the PtDAs. The comments were incorporated in the questions in R2.

Finally, in response to the question on whether the dialogues in each step represent their objective, most participants agreed ('Team talk' step, n=10; 'option talk' step, n=7; 'Decision talk' step, n=12) and made suggestions on the wording of the dialogues. Suggestions were also made to adapt the name of the original T*hree-talk* steps to a more representative one in the screening context. All the suggestions were incorporated into R2 to be approved or rejected by the other participants.

Only one of the questions evaluating the clinical practice guide did not reach the minimum Cc established: 'Do you consider a guide that concisely summarises the SDM steps to be necessary?' (*¿Cree necesaria una guía que resuma de forma rápida las fases de la TDC?*) (Cc=75). This question was incorporated into R2. In the open questions, participants suggested changing the wording of the step 1 dialogues (n=3) and incorporating a review of communicative skills

---

**Box 1  Response to the question: are the steps based on 'Three-talk' suitable for the application of shared decision-making (SDM) in breast cancer screening? Please explain briefly.**

P3 (R1): *Yes, it shows how the health professional can implement SDM in a three-step process in a brief, practical and easy-to-read way. It describes the characteristics that differentiate each step, and specific examples of implementation in breast cancer screening.*

---

**Box 2  Response to the question: how would you improve the elements selected in the previous question?**

P7: *I think that the handbook is very long, which may reduce motivation to read it.*
P6: *Very long and it doesn't show how to use the tool.*

---

 

---

> **Box 3   Response to the question: what other content would you include in the clinical practice guide?**
>
> *P3: Provide more information or example dialogues on how to use communication skills. This last (point) if the health professionals don't have a grounding or training in active listening, motivational interviewing, empathy, reflection, etc.*
> *P10: I'd go into greater depth on relationship-building skills and give a few links to where they can find exercises to train themselves (in this).*

(box 4); the same was applied to step 2, but participants added a comment about using relative risks instead of absolute ones (n=1) (box 5).

They also proposed: eliminating the definition of SDM for step 3 in the guide (n=4), incorporating a brief clarification noting that women may also consult other people for support in making their decision (n=3) and mentioning the possibility of reversing the decision (n=4) (box 6). Between 6 and 8 people stated that they would not make any change to steps 1, 2 or 3.

Finally, for the last question: 'What other content would you include in the clinical practice guide?', participants reiterated the need to include a review of communication skills (n=3) and one of them proposed changing the self-assessment to use either the ASQ3 or the CollaboRATE instrument.

### Round 2

R2 was structured around/was based on open-question responses and included the elements about which agreement had not been reached in the previous round. Thirteen Likert scale questions, 5 multiple-choice questions and 6 open questions were produced in the handbook. For the clinical practice guide, two Likert scale questions and five open questions were included (table 3).

Of the 13 Likert Scale questions, only three reached a score of Cc >75. These underlined the need to: reduce the length of the handbook (Cc=81.3), create a clinical practice guide to accompany the handbook (Cc=81.3) and mention the possibility of reversing the decision in the follow-up plan (Cc=87.6).

The close-ended questions included the following: 'Which elements of the handbook would you shorten?' (*¿Qué elementos reducirían del Manual?*), to which the two most significant answers were 'the Introduction' (50%) and 'None' (31.3%). Following the comments made in the previous round, alternative formulations of the sample phrases for the dialogues in each of *the Three-talk* steps were given, as well as a change of name for step 2: 'Option talk and exploring preferences'

---

> **Box 4   Response to the question: what elements would you change in step 1: 'team talk'?**
>
> *P3: I'd include a few reviews, such as (on) active listening and deliberation. Perhaps using a phrase like 'Remember to pay close attention and give assertive responses (active listening), and to think the options through carefully for the decision (deliberation)'.*

---

> **Box 5   Response to the question: What elements would you change in step 2: 'Option talk'?**
>
> *P15: Change relative risks to absolute risks.*

(*Plantear opciones y explorar preferencias*), on which consensus was reached (81,3%).

In their responses to the open questions, those who considered the proposed dialogues unrepresentative of the steps had the opportunity to suggest a re-wording. Finally, participants were able to include their final comments in both the handbook and the guide (figure 3).

Most had no further suggestions for each document, but some participants included comments about shortening the handbook (box 7) and including this material in clinical practice guides, in order to improve implementation (box 8).

### Round 3

R3 was structured according to the 10 elements on which no agreement was reached in R2. Six questions with close-ended, dichotomous answers were posed in the section evaluating the handbook, and one in the section evaluating the clinical practice guide; in addition to an open question. Of these, only those proposing an improvement to the organisation of the clinical practice guide, a change of colours and a review of cross-cutting communication skills in SDM reached a Cc of over 75% (table 4).

Since agreement was not reached on the flow diagram for the early detection of breast cancer programme, this figure was removed from the handbook, in light of the fact that it only applies to the region of Catalonia. The other elements in which no agreement was reached were the need for incorporating more samples of professional dialogues (64.7%); incorporating information about joint responsibility for the decision (41.2%); adding information on the limitations of the SDM model (58.8%), as well as adding supplementary resources on the way to use the PtDAs (52.9%) and on communication skills and competencies (58.8%). The researchers believed that the additional content would not entail substantial changes to the handbook but would provide more information for professionals who are not familiar with the model, and that is why all these elements were incorporated into the handbook.

The texts included were developed according to the proposals submitted by the participants in the previous rounds. For example, the following elements were highlighted in the professional dialogues: the possibility of reversing the decision, needing more time and accessing

---

> **Box 6   Response to the question: what elements would you change in step 3: 'decision talk'?**
>
> *P11: I'd add the possibility of reversing the decision; I'd take out the explanation about SDM.*

---

**Table 3** R2 responses

| Section | Questions using a Likert scale of 1 (completely disagree) to 6 (completely agree) | 1 | 2 | 3 | 4 | 5 | 6 | CC |
|---|---|---|---|---|---|---|---|---|
| Evaluation of the handbook on Shared Decision-making in breast cancer screening | 1. Eliminate (): Flow diagram of the early detection of breast cancer programme (page 27) | 6.3 | 18.8 | 18.8 | 6.3 | 43.8 | 6.3 | 56.4 |
| | 2. Shorten content: the handbook format is very long | 0 | 12.5 | 6.3 | 18.8 | 25 | 37.5 | 81.3 |
| | 4. Incorporate more examples of dialogues between the professional and the woman into each phase | 18.8 | 6.3 | 6.3 | 31.3 | 31.3 | 6.3 | 68.9 |
| | 5. Add information on communication skills and competency resources | 0 | 12.5 | 25 | 12.5 | 43.8 | 6.3 | 62.6 |
| | 6. Add information on joint responsibility for the shared decision-making agreement | 6.3 | 31.3 | 12.5 | 0 | 37.5 | 12.5 | 50 |
| | 7. Add information about resources on using the Patient Decision aids (PtDAs). Note that this tool is intended to be used with the women | 0 | 18.8 | 12.5 | 18.8 | 18.8 | 31.3 | 68.9 |
| | 8. Add information on the limitations of the SDM model | 6.3 | 18.8 | 25 | 12.5 | 25 | 12.5 | 50 |
| | 9. Provide sample dialogues on exploring the women's values, beliefs and preferences | 0 | 18.8 | 12.5 | 18.8 | 31.3 | 18.8 | 68.9 |

| Closed questions | Options | Percentage (%) |
|---|---|---|
| 3. Which element of the handbook would you shorten? | (a) Objective of the material | 0 |
| | (b) Who is it aimed at? | 0 |
| | (c) Introduction | 50 |
| | (d) SDM: What is it? | 0 |
| | (e) SDM: Why is it important? | 0 |
| | (f) SDM: What skills or competencies do health professionals need? | 0 |
| | (g) SDM: What do patients think? | 6.3 |
| | (h) SDM in breast cancer screening: The screening programme | 6.3 |
| | (i) SDM in breast cancer screening: Implementation of SDM in breast cancer screening | 6.3 |
| | (j) SDM in breast cancer screening: Self-assessment of the SDM process | 0 |
| | (k) None | 31.3 |
| | Total | 100 |
| 10. Change the name of phase II | (a) Option talk (current name) | 18.8 |
| | (b) Option talk and exploring preferences (proposal) | 81.3 |
| | (c) Other | 0 |
| | Total | 100 |
| 12. Phase I dialogue: Team Talk (page 34): | (a) Now that we know that you can decide what to do about screening, we're going to talk about the characteristics of screening, so that you know what your options are (current dialogue) | 12.5 |
| | (a) You have the option of deciding whether or not to have early detection tests for breast cancer. If you're happy to, we can explore together what risks and benefits the test involves for you (proposal) | 81.3 |
| | (c) Other | 6.2 |
| | Total | 100 |

**Table 3** Continued

| Section | Questions using a Likert scale of 1 (completely disagree) to 6 (completely agree) | 1 | 2 | 3 | 4 | 5 | 6 | CC |
|---|---|---|---|---|---|---|---|---|
| | 14. Phase II dialogue: Option Talk (page 36) | | | | | | | |
| | (a) I appreciate you sharing your views with me and I'm here to help you come to a good decision. Let's do a recap of your preferences and check whether you have any more questions (current dialogue) | | | | | | 18.8 | |
| | (b) I'm here to help you make a decision. Let's look at what your preferences are and the various options available, and we'll check whether you have any questions about them (proposal) | | | | | | 75 | |
| | (c) Other | | | | | | 6.2 | |
| | Total | | | | | | 100 | |
| | 16. Phase III dialogue: Decision Talk (page 38): | | | | | | | |
| | (a) Do you think that you're ready to make the decision or do you need more time? (current dialogue) | | | | | | 12.5 | |
| | (b) Now that we've gone over the advantages and disadvantages of early detection, do you think that you can make the decision? Bear in mind that this can be delayed if you need more time or to talk about it with someone of your choice (proposal) | | | | | | 81.3 | |
| | (c) Other | | | | | | 6.2 | |
| | Total | | | | | | 100 | |
| Evaluation of the Clinical Practice Guide: implementation of SDM for healthcare professionals | Questions using a Likert Scale of 1 (completely disagree) to 6 (completely agree) | 1 | 2 | 3 | 4 | 5 | 6 | Cc* |
| | 1. A clinical practice guide is necessary for this handbook | 6.3 | 0 | 12.5 | 25 | 31.3 | 25 | 81.3 |
| | 2. Improve the design of the clinical practice guide to improve understanding (colour, structure, etc) | 6.3 | 0 | 18.8 | 37.5 | 25 | 12.5 | 75 |
| | 3. Eliminate additional information (definitions of risk factors, mammography, SDM) | 6.3 | 18.8 | 18.8 | 6.3 | 18.8 | 31.3 | 56.4 |
| | 4. Mention the possibility of reversing the decision in the follow-up plan | 6.3 | 0 | 6.3 | 18.8 | 25 | 43.8 | 87.6 |
| | 5. Mention relationship-building competencies: active listening, showing empathy, clarification and so on | 12.5 | 6.3 | 12.5 | 6.3 | 43.8 | 18.8 | 68.9 |

SDM, shared decision-making.

**Figure 3** Changes made to the guide.

support from a third person to make the decision (figure 4).

The Delphi round was brought to a close in R3, taking into account the criteria cited by Martínez-Piñeiro, regarding the elements on which agreement was not reached:[23] a) The limited number of items for which Cc >75 was not achieved (6 of the 61 Likert Scale and closed questions; b) Limited resources and time; c) The possibility that participants would abandon the study in a subsequent round, which would affect the external validity of the study. The last two criteria were applied in the context of the COVID-19 pandemic, given that half of the participants were health professionals who work in health centres.

## DISCUSSION

The literature mentions certain barriers to applying SDM in breast cancer screening, including limited time in clinical appointments and health professionals' lack of training in providing more participatory care.[21] This was the motivation for producing the first handbook and clinical practice guide on this subject, aimed at supporting health professionals by providing them with the essential elements for implementing SDM among women for breast cancer screening.

The most relevant results included the validation of usefulness and relevance of support materials when using the Delphi technique, considering the experts' opinion

to reach agreements on editing the design and content, as well as their recommendation to incorporate these materials into the clinical practice guide. The Delphi technique may be adapted to a generic model—the *Three-talk model*—to one specifically designed for breast cancer screening.

Of the 43 participants who were invited to respond to the Delphi questionnaires, more than half showed interest in the topic of the research and collaborated in it. However, only 20 of them continued to participate in the study. This may be related to the time of the questionnaires, which coincided with the end of the first wave of the COVID-19 pandemic and the resurgence of cases at the beginning of the second wave. Despite this, the professionals who decided to participate at the beginning of the process fulfilled their commitment, illustrated by the fact that the number of participants simply decreased by 3 between rounds, these having been lost from the subjects category (n=3).

### Discussion between the participants

It was easy to reach an agreement on the main content of the elements in the first round. Regarding the structure and development of SDM using the T*hree-talk* model,[17] which was considered suitable for breast cancer screening, one of the participants initially suggested using the model created by the Agency for Healthcare Research and

---

**Box 7  Response to 'provide your final comments on the Handbook'**

P10: *None, the idea of including appendices on communication skills for the health professional, and on the screening tests for the women, seems like an excellent idea to me, to avoid making the handbook longer but offer additional tools for those health workers and women who would like more information.*

---

**Box 8  Response to 'provide your final comments on the guide'**

P10: *Clinical practice guidelines on the preventive approach to breast cancer that includes these points on shared decision-making would be very useful to support implementation. In any case, I don't think that it is a prerequisite to be able to produce the handbook that you are working on. This handbook could be incorporated into future Clinical Practice Guidelines (CPG).*

---

**Table 4** R3 responses

| Section | Closed questions | Options | Percentage (%) |
|---|---|---|---|
| Evaluation of the Handbook on Shared Decision-Making in Breast Cancer Screening | 1.Given that no consensus has been reached (56.4 %) on whether or not to eliminate (): Flow diagram of the early detection of breast cancer programme (page 27), please select one of the following options: | (a) Eliminate. It does not add relevant information to this handbook | 47.1 |
| | | (b) Keep. Translate to Spanish and improve the image resolution | 52.9 |
| | | Total | |
| | 2. Given that there is no consensus (68.9 %) about whether to add more examples of dialogues between the professional and the women for each phase, please select one of the following options: | (a) One example per phase (current format) | 35.3 |
| | | (b) Three examples per phase (proposed new format) The image will be adapted to a more readable size for the handbook | 64.7 |
| | | Total | 100 |
| | 3. Given that there is no consensus (62.6 %) about whether to add information on communication skills and competencies resources to the handbook, please select one of the following options: | (a) Yes, it is necessary to incorporate bibliographic references into the handbook for those who would like to find out more about this topic | 58.8 |
| | | (b) No, the handbook is already too long to add more information | 64.7 |
| | | Total | 100 |
| | 4. Given that there is no consensus (50%) about whether to include information on joint responsibility for the SDM agreement, please select one of the following options: | (a) Yes, it should be included because the information is not clear | 41.2 |
| | | (b) It is not necessary, it is already clear that the responsibility is shared | 58.8 |
| | | Total | 100 |
| | 5. Given that there is no consensus (68.9%) about whether bibliographic references should be added on the PtDAs—note that the PtDAs is an appendix to the handbook, to be used by the woman and health professional—please select one of the following options: | (a) Yes, they should be added | 52.9 |
| | | (b) No, this is not necessary | 47.1 |
| | | Total | 100 |
| | 6. Given that there is no consensus (50%) about whether to add information on the limitations of the model, please select one of the following options: | (a) Yes, this is necessary because not doing so would mean producing one-sided material | 58.8 |
| | | (b) No, it is not necessary because the objective of the handbook is to show the advantages of implementing it | 41.2 |
| | | Total | 100 |
| Evaluation of the Clinical Practice Guide: Implementation of SDM for Health Professionals | 1. Given that there is no consensus about the design and content of the guide, please select one of the following options. The infographic will be adapted to a more readable size for the guide | (a) Current format | 23.5 |
| | | (b) Proposed new format | 76.5 |
| | | Total | 100 |

PtDAs, Patient Decision aids; SDM, shared decision-making.

Quality.[24] However, this alternative model contains five steps, and the model proposed by the authors, with fewer steps, met all the requirements of SDM. Regarding the set of nine figures in the handbook, only one was eliminated, and the wording of three was edited.

The participants easily agreed that the initial version of the handbook was very long (56 pages). Its length was due to the fact that it would be published in a pocket edition, which corresponds to 23 pages in a larger textbook edition. The researchers decided to maintain the smaller format because it is more transportable, whereas they eliminated the content elements agreed by the participants.

It was impossible to reach an agreement on six items. While agreement should be ideally reached for all items, yet, when a new round does not provide more information or it is unlikely to achieve a better result, the rounds of questions may come to an end despite there being a small number of disagreements remaining.[21] The formulation changes of the responses between R2 (Likert scale) and R3 (dichotomous) meant that participants had to choose one of the options rather than rate their level of agreement on the statements, which undoubtedly made it more difficult to reach an agreement.

Certain responses to the open questions were analysed in-depth by the researchers. One of the participants in R1 suggested that the professional self-assessment method could be changed from SDM-Q-doc[25] to Ask 3Q[26] or CollaboRATE.[27] However, Ask 3Q is a methodology for applying SDM, making it equivalent to the Three-talk model. Given that the Three-talk model received positive evaluation from the participants, the change was not made. The other tool, CollaboRATE, is designed for the patient's evaluation of the professional, which was not the purpose of this questionnaire.[28] Our objective was to enable the professional to evaluate the way he or she performs SDM, resulting in a self-guided learning of this methodology. The researchers, therefore, kept the original version, SDM-Q-doc, and adapted it for screening.

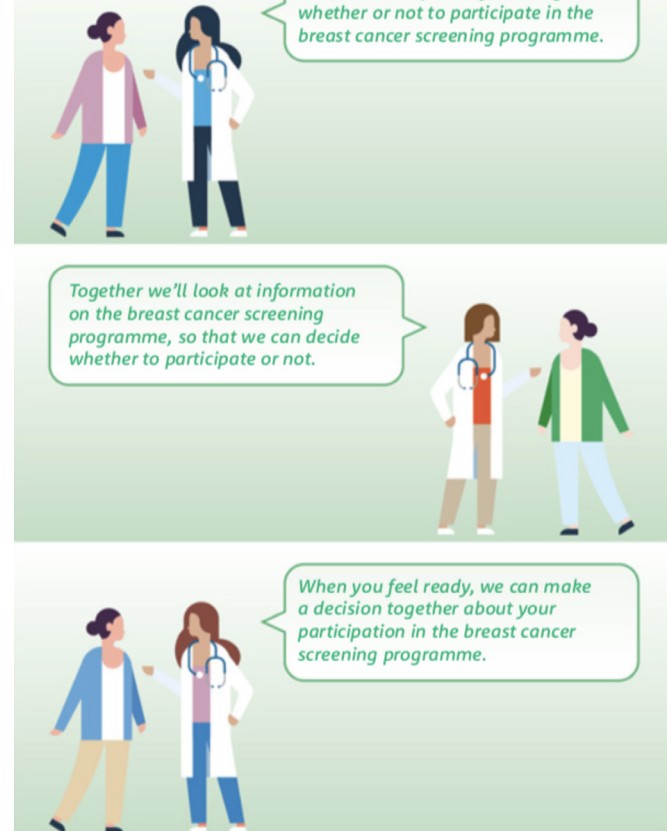

**Figure 4** Example of dialogues for the professionals in the 'team talk' step.

The decision on the flow diagram was affected by whether participants came from the region of Catalonia (of those living in Catalonia, 5/6 wanted to keep it, although improving its resolution; in contrast, the specialists from outside Spain (7/11) opted to remove it). Given that the objective of the handbook is to be used in other territories, the research group decided to eliminate the flow diagram.

The sample dialogues suggesting how professionals should conduct SDM at each point in the process were widely accepted as a fundamental part of the handbook, even though no consensus was reached on whether to include more sample dialogues for each step (Su=4/6; Sp=7/11). While Cc >75 was not reached, a larger proportion of both groups advocated for providing more examples. This may be directly related to the fact that both groups believed that SDM training for health professionals is still incomplete. Some of these participants therefore called for the handbook to provide more support, giving professionals greater confidence in implementation using the dialogues. The same conclusion can be reached regarding the decision to include more bibliographic references on communication skills and relationship-building competencies (Su=3/6; Sp=7/11) and including information about PtDAs (Su=5/6; Sp=4/11). In the latter case, the results differed from the two groups: most

of the Subject participants wanted to add information to these tools, perhaps highlighting their lack of knowledge about them or lack of training in their use, while Specialists did not consider their inclusion so relevant, due to their familiarity with the tools.

### How to improve the application of SDM to screening

While 83% of health professionals were strongly interested in promoting SDM during the clinical encounter,[28] they admitted that their lack of training in the SDM model was one of the most significant barriers to its implementation in the screening context.[13]

A review of the training health professionals had received confirmed our belief that there is a lack of strategies to familiarise health professionals with this model. In Spain, the topic has been introduced into medicine and health-related degree programmes.[29–32] However, it is not framed accurately within a SDM model, although, it is closer to communication or clinical communication skills, which have been used interchangeably as equivalents to the model. The level of accuracy and strategies used in this training are also unknown. Most training in SDM is acquired in postgraduate-level studies aimed for doctors and nurses,[33] whereas particular attention should also be paid to health workers in primary care centres (including support and technical staff, as well as clinicians), who provide person-centred healthcare in a holistic manner.[34]

Experts in SDM have argued it is necessary to prioritise adapting curricula to consolidate this training, by emphasising an education in communication skills and the accreditation of these competencies,[35] within the framework of a horizontal care model. In addition, experts highlight the need to create partnerships between universities and interdisciplinary research groups to develop this material.[35]

Experts also recommend a training methodology based on practical activities such as role plays, as well as teamwork, in teams of six people for instance, in a day-long training, and providing constructive feedback on students' capacity to express empathy, giving assertive responses, engaging them in active listening, and other skills.[36] This handbook and clinical practice guide, therefore, includes dialogues and specific examples of how to apply them. And it will serve as reference material supporting an initial grounding in SDM for professionals who have not received any formal training in this subject, but also as supplementary material for those who have; enabling them to apply the skills and competencies acquired in the specific context of breast cancer screening.

The final structure of our document responds to the need described in the preceding paragraph and is highlighted by the participants in the study.

Given the change of paradigm that SDM entails, all measures that help familiarise professionals with SDM are important. For example, adding a section into the clinical practice guidelines on how to include the patient in decision-making; thereby, coordinating evidence-based practice with SDM,[37] may be useful. Patients may even

participate, to some extent, in its development, as it is a current practice in such organisations such as the National Institute for Health and Care Excellence and the Scottish Intercollegiate Guidelines Network.[38] In this sense, our proposed handbook and clinical practice guide, as well as the PtDAs, whose quality has already been evaluated and certified by international organisations such as The Ottawa Hospital,[39] may be considered as complementary materials.

## LIMITATIONS

The main limitation of the study was participant recruitment, which is a typical constraint. It was a particular problem in this case, since the empirical work coincided with the successive waves of the COVID-19 pandemic, which hindered the active participation of some professionals who had initially agreed to participate in the study. Despite this, there were fewer withdrawals round 2 onwards that might have been expected in those circumstances.

The change in the formulation of the R2 (Likert Scale) and R3 (dichotomous) responses may have made it more difficult to reach the established minimum Cc for agreement. Nevertheless, with reference to Martínez-Piñeiro[23] and Benner et al,[23 21] the research team determined that one more round would not have provided any added value to the results, as seen in the reasons described in the preceding sections. Nevertheless, the decision made regarding those elements about which no agreement had been reached did not significantly affect the participants' opinions regarding the basic concepts on which the initial questionnaire was based.

Finally, it should be noted that a systematic literature review (2018) showed that further research is still needed to determine the real impact that training interventions have on health professionals regarding SDM, since the level of certainty of the studies was low or very low. In this research, professionals who had received standard training were compared with those who had been trained in SDM; from the 15 studies, it was concluded that the results for patients' satisfaction, knowledge, decision-related conflict, regret, level of health and quality of life differed little or not at all from one to another.[33] Despite this, the demand for information and training expressed by this study's participants makes us believe that this first handbook aimed at health professionals for implementation in a breast cancer screening context will help clarify the healthcare model focused on patients' needs and preferences. However, we have also noted the need to expand the training in SDM and develop empirical strategies to facilitate its implementation.

## CONCLUSION

A horizontal relationship between patients and health professionals enables person-centred care to be delivered, in which that patient is considered a protagonist in the decisions made on his or her health. This has been recognised by several governmental organisations and incorporated into discourse and strategies. However, the practical application of this model is an area in which progress is still to be made. The handbook and clinical practice guide therefore aim to familiarise professionals with the model, helping them to engage women in the decision of either having breast cancer screening or not. The results obtained enable us to conclude that, to apply it as a public policy, first there must be a pilot study with health professionals, which should be supplemented by formal training in SDM.

**Author affiliations**
[1]Economic, Universitat Rovira i Virgili, Reus, Spain
[2]Research Centre on Economics and Sustainability (ECO-SOS), Reus, Spain
[3]Research Group on Statistics, Economic Evaluation and Health (GRAEES), Reus, Spain
[4]Terrassa University School of Nursing and Occupational Therapy (EUIT), Universitat Autònoma de Barcelona, Terrassa, Spain
[5]Evaluation and Research in the Field of Social Sciences and Health, Area Q, Barcelona, Spain
[6]Cancer Prevention and Control Programme, Catalan Institute of Oncology, Hospitalet de Llobregat, Spain

**Acknowledgements** The authors thank the three expert reviewers in shared decision-making and breast cancer: Victor Montori, Lilisbeth Perestelo-Pérez and Montserrat Rué; as well as the external reviewers, Lluís Colomés Figuera and Josep Maria Sabaté. The authors also thank the 20 participants in the study for their time, effort and perseverance in answering all the rounds of questions the research team had posed them.

**Collaborators** ProShare group: Misericòrdia Carles-Lavila, Núria Codern-Bové, María José Hernández-Leal, María José Pérez, Roger Pla, Vanesa Ramos, Carmen Vidal.

**Contributors** MJHL, MJPL and MCL: designing of the work, data analysis, interpretation, manuscript writing, obtaining funding and guarantor. NC-B and AC: collaboration in design of the work, interpretation and collection of data. CVL: contribution of patients or study material. All authors critically reviewed the manuscript and approved of the final version.

**Funding** Financial support for this study was provided entirely by a grant from Instituto de Salud Carlos III through the project PI18/00773 (co-funded by the European Regional Development Fund), and by the European Union's Horizon 2020 research and innovation programme, under Marie Skłodowska-Curie grant agreement No 713679 from Universitat Rovira i Virgili (URV). The funding agreement ensured the authors' independence in designing the study, interpreting the data, and writing and publishing the report.

**Competing interests** None declared.

**Patient consent for publication** Not applicable.

**Ethics approval** This study involves human participants and was approved by the Medicinal Product Research Ethics Committee (CEIm) of the Institut d'Investigació Sanitària Pere Virgili (Pere Virgili Health Research Institute).

**Provenance and peer review** Not commissioned; externally peer reviewed.

**Data availability statement** Data are available upon reasonable request. All data relevant to the study are included in the article or uploaded as supplementary information. Contact details: misericordia.carles@urv.cat

**ORCID iD**

María José Hernández-Leal http://orcid.org/0000-0002-4002-6454

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
