## [Reviewer comments · BMJ Open]

ARTICLE DETAILS

TITLE (PROVISIONAL)	Development of support material for health professionals who are implementing Shared Decision-Making in breast cancer screening: Validation using Delphi technique
AUTHORS	Hernández Leal, María José; Codern-Bové, Núria; Pérez-Lacasta, María José; Cardona, Angels; Vidal, Carmen; Carles-Lavila, Misericòrdia

VERSION 1 – REVIEW

REVIEWER	Ivlev, Ilya Kaiser Permanente Center for Health Research Northwest Region
REVIEW RETURNED	07-Jun-2021

GENERAL COMMENTS	OVERAL My main concern is that the authors used the term “shared decision making” (SDM) when referring to the decision whether to screen in women for whom routine screening is recommended. The idea for SDM is to use this technique in cases when there is no clear net benefit. For example, the US Preventive Services Task Force recommends SDM in women with an average risk for breast cancer and 40-49 and 75+ years. Routine screening is recommended for women aged 50-74. SDM is not a technique to encourage screening. However, the authors developed an SDM educational tool to increase the screening rate among women for whom routine screening is recommended. This is a misconception of SDM that will confuse the reader. It appears to me that the developed tool has very little to do with SDM. The use of the Delphi panel seems unjustifiable because the purpose was (the best of my judgment) to evaluate the handbook. The evaluation criteria have never been stated. ABSTRACT Overall abstract: The abstract should explicitly state the methods. The purpose of the Delphi panel is unclear for the reader. The results are surprising because the authors have never stated the purpose for the panelists. It's confusing for me why the authors used the Delphi panel to evaluate a handbook. 1. The first sentence is unclear. Did you mean that no training exists or not documentation of training exists? I suggest the authors rewrite this sentence. Also, reflect the real situation “no” meant that this doesn't exist anywhere. Did you want to say this training/or documentation o training doesn't exist in Sapin? In your hospital?2. Line 8: “to this topic” – It's unclear which topic the authors are referring to.3. Objective: It's unclear if the handbook and the practice guide ais
---

	the same document. Please clarify. 4. The objective is “to evaluate the handbook” – Why did the authors use a Delphi panel? Would it be more useful if authors focus on usefulness and relevance? It’s possible that I don’t understand why you used this method because I still didn’t understand the purpose of this study. 5. Design: Please, refer to Delphi as a method or technique, not “methodology.” 6. Was it a modified Delphi? This should be clear. 7. “field of breast cancer prevention” – please state clearly what you meant. It will be confusing for the reader when you refer to the field of breast cancer screening. 8. The criteria for concordance are unclear. Seventy-five of what? Did you mean that 75% of Delphi panelists should agree on the importance? 9. The presented Likert scale is not meaningful for the reader. Please state what these numbers mean. 10. The design section should contain the dates for the whole Delphi and how the panel functioned. Was it an online panel? Did you use an app? 11. Not all provided conclusions were supported by the results. BACKGROUND Overall background: 1. The first paragraph is a description of the importance of breast cancer screening, although this manuscript is about developing a handbook to support shared decision making. I suggest the authors begin this section by addressing challenges in shared decision-making. However, the reader would also benefit from learning about breast cancer screening recommendations in Spain. The authors might consider the following questions: When is shared decision-making recommended in Spain? When is routine screening recommended? What is the current practice? How is shared decision-making implemented in Spain? The reader needs to understand current practice. 2. Suppose the authors decide to keep the description of the benefits and harms of breast cancer with mammography. In that case, they should update their references to use the current evidence. METHODS 1. The Delphi technique is not described in sufficient detail. The reader will remain to wonder what actually has been done during the study. 2. The presented handbook is in English. Were all Delphi participants versita=ile in English. Have they reviewed a Spanish version? 3. The provided methods description is not sufficient to understand the scientific rigor. Recruitment and retention don’t contain sufficient details. RESULTS The main issue with the results is that it’s unclear how the handbook changed with the Delphi rounds. Why did panelists reconsider their judgments? Perhaps, it would be helpful if the authors provided the first draft of the handbook.
--	---

REVIEWER	Bombard, Yvonne Li Ka Shing Knowledge Institute, St. Michaels' Hospital
REVIEW RETURNED	16-Jul-2021

GENERAL COMMENTS

Thank you for the opportunity to review this interesting descriptive study related to shared decision-making (SDM) for breast cancer screening. This manuscript details three rounds of Delphi panels to solicit the views of healthcare professionals on a handbook and clinical practice guide for the application of SDM in breast cancer screening. Overall, this study provides a concise introduction to the breast cancer screening landscape and needs for patients and providers and a robust results section chronicling multiple rounds of Delphi. The following is intended to provide constructive feedback.

Background: There is widespread adoption of SDM as best practice in supporting individuals making breast cancer screening decisions and numerous tools, guidelines, and frameworks are now available to operationalize SDM in clinical settings. It is somewhat unclear what novelty this study brings to the literature on this topic. The background outlines a gap in evidence in Spain, but a justification for the use of a handbook paired with a clinical practice guide to alleviate these gaps is missing. The authors do not cite strategies and modalities to address this topic from other contexts. Why is this handbook the best tool to support SDM for healthcare professionals in Spain? Would a handbook and practice guide be readily adopted within this context? Consider including an implementation science framing as part of any justification provided and mapping this evaluation component of the tool's development alongside implementation and services outcomes.

In addition to the attached appendices, a primer on the tools is needed in this section – please provide a brief description of how the tool works, and the steps of the applied Three-talk Model. Please include here a description of self-assessment component of the guide referenced in R1 results and discussion sections. The Three-talk model is positioned as the framework for the tool design, but it remains unclear how this framework informs broader study design and methods. The study objective is missing.

Methods: Without a clear objective statement in the background, the use of Delphi methodology is not fully justified or in line with the purpose stated in the abstract – to evaluate the tools created. The Delphi appears to be applied here as part of tool development and to evaluate consensus on the tool, not the tool itself. Why was Delphi selected over other evaluative methods, especially given this is a topic where extensive expert commentary exists in the literature from comparable jurisdictions? How was the sample size and proportion of subjects and specialists needed for this Delphi determined? Was there any attempt to involve subject-specialists, healthcare investigator-practitioners, who would have the relevant expertise necessitated by Delphi? Please include how data was collected under data collection – presumably Google Form as per earlier paragraph but should be inserted here.

Results: Comprehensive reporting of each round of Delphi, consensus reached, and elements carried forward to subsequent round. Please insert subject/specialist breakdown alongside the professional demographics at the top of this section. Consider expanding the flowchart to include purpose of each round of questionnaire and its timing staggered between response rates.

Discussion: Instead of removing the Catalonia specific flowchart be, could the tool be iterated for different localities with tailored specifications? What are the limitations inherent to Delphi and how were they addressed in this study?

This manuscript was reviewed with Suvetha Krishnapillai (MSc candidate), University of Toronto.

VERSION 1 – AUTHOR RESPONSE

Reviewer 1: Dr. Ilya Ivlev, Kaiser Permanente Center for Health Research Northwest Region

OVERAL	
My main concern is that the authors used the term “shared decision making” (SDM) when referring to the decision whether to screen in women for whom routine screening is recommended. The idea for SDM is to use this technique in cases when there is no clear net benefit. For example, the US Preventive Services Task Force recommends SDM in women with an average risk for breast cancer and 40-49 and 75+ years. Routine screening is recommended for women aged 50-74. SDM is not a technique to encourage screening. However, the authors developed an SDM educational tool to increase the screening rate among women for whom routine screening is recommended. This is a misconception of SDM that will confuse the reader. It appears to me that the developed tool has very little to do with SDM. The use of the Delphi panel seems unjustifiable because the purpose was (the best of my judgment) to evaluate the handbook. The evaluation criteria have never been stated.	The introduction expands the definition of SDM as its objective and explains that it is not intended to increase adherence, however this is a secondary consequence in certain cases (page 3). The objective of the materials is written again, highlighting that this is TRAINING FOR PROFESSIONALS who want to apply the SDM in women who must decide whether or not to perform screening. While Annex 1 is a complementary PtDAs to carry out the SDM. The criteria are mentioned in the methodology and the formula how the Concordance Coefficient is reached in each question, which is considered as an indicator of consensus (page 5 “Data Analysis” in highlight).
ABSTRACT	
Overall abstract: The abstract should explicitly state the methods. The purpose of the Delphi panel is unclear for the reader. The results are surprising because the authors have never stated the purpose for the panelists. It’s confusing for me why the authors used the Delphi panel to evaluate a handbook.  1. The first sentence is unclear. Did you mean that no training exists or not documentation of training exists? I suggest the authors rewrite this sentence. Also, reflect the real situation “no” meat that this doesn’t exist anywhere. Did you want to say this training/or documentation o training doesn’t exist in Spain? In your hospital? 2. Line 8: “to this topic” – It’s unclear which topic the authors are referring to. 3. Objective: It’s unclear if the handbook and the practice guide ais the same document. Please clarify. 4. The objective is “to evaluate the handbook” – Why did the authors use a Delphi panel? Would it be more useful if authors focus on usefulness and relevance? 	The entire section was rewritten, among them the objective was changed being the center the usefulness and relevance of the material (manual and guide) for health professionals. The word "methodology" was changed to "technical" Delphi throughout the document (including the title). Likert scale scores were specified. 75 is a match coefficient between participants for each question, not a percentage. Added formula and explanation is in “Data analysis” (highlight). Study dates were added to the summary. Added the Delphi platform (Google form). The conclusions left only the predominant outcomes of the Delphi, including utility and limitations.

It's possible that I don't understand why you used this method because I still didn't understand the purpose of this study. 5. Design: Please, refer to Delphi as a method or technique, not "methodology." 6. Was it a modified Delphi? This should be clear. 7. "field of breast cancer prevention" – please state clearly what you meant. It will be confusing for the reader when you refer to the field of breast cancer screening. 8. The criteria for concordance are unclear. Seventy-five of what? Did you mean that 75% of Delphi panelists should agree on the importance? 9. The presented Likert scale is not meaningful for the reader. Please state what these numbers mean. 10. The design section should contain the dates for the whole Delphi and how the panel functioned. Was it an online panel? Did you use an app? 11. Not all provided conclusions were supported by the results.	
BACKGROUND	
1. The first paragraph is a description the importance of breast cancer screening, although this manuscript is about developing a handbook to support shared decision making. I suggest the authors begin this section by addressing challenges in shared decision-making. However, the reader would also benefit from learning about breast cancer screening recommendations in Spain. The authors might consider the following questions: When is shared decision-making recommended in Spain? When is routine screening recommended? What is the current practice? How is shared decision-making implemented in Spain? The reader needs to understand current practice. 2. Suppose the authors decide to keep the description of the benefits and harms of breast cancer with mammography. In that case, they should update their references to use the current evidence.	The introduction was redrafted starting with the definition and implications of the SDM (Paragraph 1) and the recommendations for breast cancer screening were eliminated (so these references were not updated) and only the current lines were maintained in Catalonia-Spain. The current status of SDM in breast cancer screening in Spain was added in paragraph 2 (includes current practice and recommendations). "In Spain there is no evidence that explicitly recommends in which type of women SDM should be performed, however its use through PtDAs has been extended to all women who must decide to have mammography because they are integrated into breast cancer screening programmes in each Autonomous Community. However, PtDAs are not commonly used in clinical appointments (5)."
METHODS	
1. The Delphi technique is not described in sufficient detail. The reader will remain to wonder what actually has been done during the study. 2. The presented handbook is in English. Were all Delphi participants versita=ile in English. Have they reviewed a Spanish version? 3. The provided methods description is not sufficient to understand the scientific	More describe the technique in "Delphi technique" (page 4). It included: objectives, requirements, process and some limitations were added. Attached material in Spanish (original with which the Delphi was made) (Annexes 2,3,4,5). The methodology was restructured,

rigor. Recruitment and retention don't contain sufficient details.	describing in more detail the step by step in the "Procedure and Data collection" section (page 5) "The researcher use two sampling strategies were used to recruit participants: convenience sampling for the specialists and snowball sampling for the health professionals. For the specialist, the researcher looking for published articles about SDM and contact the authors by e-mail (MJH, MC, MJP). For the health professionals, the researchers sent an e-mail to invitations (NC, AC), and they could be resent to other collages. Finally, researchers (NC-AC) sent invitations by email to 43 potentials experts for participate in a Delphi, 30 whom accepted. The aim was determinate the usefulness of the topics, relevance of the content and document design of the material for the SDM on BC screening. The Delphi was making on Google form between July and October 2020."
RESULTS	
The main issue with the results is that it's unclear how the handbook changed with the Delphi rounds. Why did panelists reconsider their judgments? Perhaps, it would be helpful if the authors provided the first draft of the handbook.	The initial (Annexes 4 and 5) and final version (Annexes 2 and 3) of handbook and guide in Spanish is attached. In addition, figures with before and after the agreed changes were added to the results (Figure 2, 3 and 4).

Reviewer 2: Dr. Yvonne Bombard, Li Ka Shing Knowledge Institute, St. Michaels' Hospital, University of Toronto Institute of Health Policy Management and Evaluation

Comments to the Authors	
Thank you for the opportunity to review this interesting descriptive study related to shared decision-making (SDM) for breast cancer screening. This manuscript details three rounds of Delphi panels to solicit the views of healthcare professionals on a handbook and clinical practice guide for the application of SDM in breast cancer screening. Overall, this study provides a concise introduction to the breast cancer screening landscape and needs for patients and providers and a robust results section chronicling multiple rounds of Delphi. The following is intended to provide constructive feedback.	Thank you for your feedback and your interest in reviewing this article. Changes have been made considering the comments of the two reviewers. However, changes may be brief at times because of following the recommendations of the article's length.
BACKGROUND	
There is widespread adoption of SDM as best practice in supporting individuals making breast cancer screening decisions and numerous tools, guidelines, and frameworks are now available to operationalize SDM in clinical settings. It is somewhat unclear what novelty this study	A paragraph was added explaining why it was decided to create a manual for this topic (page 3). "A training material are the manuals, as these are a useful tool to transmit knowledge and provide quick and simple information on how to operationalize new practices, turning

brings to the literature on this topic. The background outlines a gap in evidence in Spain, but a justification for the use of a handbook paired with a clinical practice guide to alleviate these gaps is missing. The authors do not cite strategies and modalities to address this topic from other contexts. Why is this handbook the best tool to support SDM for healthcare professionals in Spain? Would a handbook and practice guide be readily adopted within this context? Consider including an implementation science framing as part of any justification provided and mapping this evaluation component of the tool's development alongside implementation and services outcomes. In addition to the attached appendices, a primer on the tools is needed in this section – please provide a brief description of how the tool works, and the steps of the applied Three-talk Model. Please include here a description of self-assessment component of the guide referenced in R1 results and discussion sections. The Three-talk model is positioned as the framework for the tool design, but it remains unclear how this framework informs broader study design and methods. The study objective is missing.	novices on a theme into an advanced users in order to using it as they use it (12). Considering that SDM is not a common practice, a manual could to some extent fill knowledge gaps on this model.” The purpose of this study was incorporated at the end of the paragraph. “The objective of this study is the experts to determine the usefulness and relevance of the contents and design provided in a handbook -manual- and a guide to support the application of Shared Decision-Making to breast cancer screening intended to healthcare professionals” The use of the manual, the self-assessment and the guide was briefly described, as well as their usefulness. “These documents should be used as reference material by health professionals when facing the decision with women on -or not- to perform mammography, taking into consideration key elements are providing the patient with information and education, interpersonal communication between doctor and patient, finally a decision (14). To facilitate the implementation of SDM the model was used as a reference the Three-talk model, adapting sus three steps to BC screening: 1) Team talk; 2) Option talk and exploring preferences; 3) Decision talk (15). A self-assessment of the SDM was included in the manual, which should be applied at the end of the appointment so professionals can identify strengths and weaknesses in the implementation of the SDM. Finally, the guide provides a summary of the handbook to be used in the same appointment as a reminder of the three steps”
METHODS	
Without a clear objective statement in the background, the use of Delphi methodology is not fully justified or in line with the purpose stated in the abstract – to evaluate the tools created. The Delphi appears to be applied here as part of tool development and to evaluate consensus on the tool, not the tool itself. Why was Delphi selected over other evaluative methods, especially given this is a topic where extensive expert commentary exists in the literature from comparable jurisdictions? How was the sample size and proportion of subjects and specialists needed for this Delphi determined? Was there any attempt to involve subject-specialists, healthcare investigator-practitioners, who would have the relevant expertise necessitated by Delphi? Please include how data was collected under data	It was incorporated in the methodology more detail of the technique Delphi. Which is more in tune with the new objective proposed. In addition, information was incorporated from a study that comments on when the use of Delphi is useful, including for subjects that are little studied and that try to implement competencies for health professionals (page 4). The sample size was made according to the literature as the optimal number of professionals (page 4; section Participants). “To determine the size of the sample, literature was consulted, which mentions that large numbers (more than 50 people) could imply an impediment in so many people reaching agreement in a limited time. Moreover, it depends on the heterogeneity of

collection – presumably Google Form as per earlier paragraph but should be inserted here.	the experts, if they are of various subjects and international enrich the opinions formulated (18). Therefore, a limit between 7 and 30 (18) was decided, with the most common being a total of 15 to 20 experts (18).” Added in "Procedure and Data collection" more detail of the step by step with which the Delphi was made, including the data collection platform: Google form. “The researcher use two sampling strategies were used to recruit participants: convenience sampling for the specialists and snowball sampling for the health professionals. For the specialist, the researcher looking for published articles about SDM and contact the authors by e-mail (MJH, MC, MJP). For the health professionals, the researchers sent an e-mail to invitations (NC, AC), and they could be resent to other collages. Finally, researchers (NC-AC) sent invitations by email to 43 potentials experts for participate in a Delphi, 30 whom accepted. The aim was determinate the usefulness of the topics, relevance of the content and document design of the material for the SDM on BC screening. The Delphi was making on Google form between July and October 2020.”
RESULTS	
Comprehensive reporting of each round of Delphi, consensus reached, and elements carried forward to subsequent round. Please insert subject/specialist breakdown alongside the professional demographics at the top of this section. Consider expanding the flowchart to include purpose of each round of questionnaire and its timing staggered between response rates.	Table 1 shows the characteristics of the participants. The goal of each Round is incorporated into the flow chart (figure 1). To better understand the changes, a before and after figures and the end of the manual were incorporated (Figure 2,3,4). Also attached are both Spanish versions, as this was the language in which the Delphi was developed and with which agreements were reached. (Annexes 4, 5).
DISCUSSION	
Instead of removing the Catalonia specific flowchart be, could the tool be iterated for different localities with tailored specifications? What are the limitations inherent to Delphi and how were they addressed in this study?	Because the flujogram describes how the screening is done in a specific section of Tarragona, it was also decided to eliminate so that the Manual is useful in several territories of Catalonia. The limitations of Delphi are mentioned in 1) The methodology (page 4) which is also mentioned in the conclusions: other methodologies are required to validate the material, such as a pilot where documents are used in clinical practice. “One of the limitation Delphi techniques is that this provides expert opinion, but could also be

	considered other complementary techniques to determine a final position on the subject of study (16-18)” 2) The other limitations of the Delphi are in the section "Limitations" when there is no consensus what to do? In this case, Martínez's criteria for closing rounds was considered. (Page 12). “The change in the formulation of the R2 (Likert scale) and R3 (dichotomous) responses may have made it more difficult to reach the established minimum Cc for agreement. Nevertheless, with reference to Martínez (2003) (19), the research team determined that one more round would not have provided any added value to the results, for the reasons described in the preceding sections. Nevertheless, the decision made regarding those elements about which no agreement had been reached did not significantly affect the participants' opinions regarding the basic concepts on which the initial questionnaire was based. “
--	--

Fecha de envío: 03-Oct-2021

bmjopen-2021-052566 - "Development of support material for health professionals who are implementing Shared Decision-Making in breast cancer screening: Validation using Delphi technique"

Comentarios del editor

COMENTARIOS	
Por favor proporcione números de página junto a cada ítem en la lista de verificación de SQUIRE para que puedan ser encontrados en el manuscrito. Por favor, no deje espacios en blanco e indique cualquier elemento que no se aplique a su diseño de estudio como 'No Aplicable'.	

Reviewer 1: Dr. Ilya Ivlev, Kaiser Permanente Center for Health Research Northwest Region

GENERAL	
Mi principal preocupación es que los autores utilizaron el término “toma de decisiones compartida” (SDM, por sus	En la introducción se amplía la definición de SDM cual es su

siglas en inglés) al referirse a la decisión de realizar un cribado en mujeres para las que se recomienda el cribado de rutina. La idea de SDM es utilizar esta técnica en los casos en que no hay un beneficio neto claro. Por ejemplo, el Grupo de Trabajo de Servicios Preventivos de EE. UU. Recomienda el SDM en mujeres con un riesgo promedio de cáncer de mama y de 40 a 49 años y más de 75 años. Se recomiendan exámenes de detección de rutina para mujeres de 50 a 74 años. El SDM no es una técnica para fomentar la detección. Sin embargo, los autores desarrollaron una herramienta educativa SDM para aumentar la tasa de detección entre las mujeres para las que se recomienda la detección de rutina. Ete es un concepto erróneo de SDM que confundirá al lector. Me parece que la herramienta desarrollada tiene muy poco que ver con SDM. El uso del panel Delphi parece injustificable porque el propósito era (según mi criterio) evaluar el manual. Los criterios de evaluación nunca se han establecido.	objetivo y se explica que esta no tiene la finalidad de aumentar la adherencia, sin embargo esta es una consecuencia secundaria en ciertas ocasiones (página 3). Se escribe nuevamente el objetivo de los materiales, resaltando que este es de ENTRENAMIENTO PARA PROFESIONALES que quieran aplicar la SDM en mujeres que deben decidir de realizarse o no el screening. Mientras que el Anexo 1 es una PtDAs -complementaria al material-para llevar a cabo la SDM Los criterios están mencionados en la metodología y la fórmula cómo se llega al Coeficiente de Concordancia en cada pregunta, el cual se considera como indicador de consenso (pagina 5 en amarillo).
RESUMEN GENERAL	
El resumen debe indicar explícitamente los métodos. El propósito del panel Delphi no está claro para el lector. Los resultados son sorprendentes porque los autores nunca han declarado el propósito de los panelistas. Me resulta confuso por qué los autores utilizaron el panel Delphi para evaluar un manual. 1. La primera oración no está clara. ¿Quiso decir que no existe formación o que no existe documentación de formación? Sugiero que los autores reescriban esta oración. Además, refleja la situación real que esto no existe en ningún lado. ¿Quería decir que esta formación / o documentación o formación no existe en España? ¿En tu hospital? 2. Línea 8: "a este tema": no está claro a qué tema se refieren los autores. 3. Objetivo: no está claro si el manual y la guía práctica son el mismo documento. Por favor aclare. 4. El objetivo es "evaluar el manual" - ¿Por qué los autores utilizaron un panel Delphi? ¿Sería más útil que los autores se centraran en la utilidad y la relevancia? Es posible que no entienda por qué usó este método porque todavía no entendí el propósito de este estudio. 5. Diseño: Por favor, refiérase a Delphi como un método o técnica, no como una "metodología". 6. ¿Fue un Delphi modificado? Esto debería quedar claro. 7. "field of breast cancer prevention": indique claramente lo que quiso decir. Será confuso para el lector cuando se refiera al campo de la detección del cáncer de mama. 8. Los criterios de concordancia no están claros. ¿Setenta y cinco de qué? ¿Quiso decir que el 75% de los panelistas de Delhi deberían estar de acuerdo con la importancia? 9. La escala de Likert presentada no es significativa para el lector. Indique qué significan estos números. 10. La sección de diseño debe contener las fechas de todo Delphi y cómo funcionó el panel. ¿Fue un panel en línea? ¿Usaste una aplicación? 11. No todas las conclusiones proporcionadas fueron	Se redactó nuevamente la sección completa, entre ellas se cambió el objetivo siendo el centro la utilidad y la relevancia del material (manual y guía) para los profesionales de salud. Se cambió de método a técnica Delphi en todo el documento Se precisó los puntajes de la escala Likert. El 75 es un coeficiente de concordancia entre los participantes por cada pregunta, no representa un porcentaje. Se agregó la fórmula y su explicación está en Data analysis (amarillo) Se agregó las fechas del estudio en el resumen. Se agregó la plataforma en que se realizó el Delphi. En las conclusiones se dejaron solo los resultados predominantes del Delphi, entre ellos la utilidad y limitaciones.

respaldadas por los resultados.	
BACKGROUND	
1. El primer párrafo es una descripción de la importancia de la detección del cáncer de mama, aunque este manuscrito trata sobre el desarrollo de un manual para apoyar la toma de decisiones compartida. Sugiero que los autores comiencen esta sección abordando los desafíos en la toma de decisiones compartida. Sin embargo, el lector también se beneficiaría de conocer las recomendaciones de detección del cáncer de mama en España. Los autores podrían plantearse las siguientes cuestiones: ¿Cuándo se recomienda la toma de decisiones compartida en España? ¿Cuándo se recomiendan las pruebas de detección de rutina? ¿Cuál es la práctica actual? ¿Cómo se implementa la toma de decisiones compartida en España? El lector debe comprender la práctica actual.	

VERSION 2 – REVIEW

REVIEWER	Bombard, Yvonne Li Ka Shing Knowledge Institute, St. Michaels' Hospital
REVIEW RETURNED	29-Nov-2021

GENERAL COMMENTS	Thank you for considering the reviewer feedback. We appreciate the thoughtfulness taken to add clarifications and revisions. There remain some points of concern with the study that haven't been sufficiently addressed. In addition, the manuscript needs careful copy-editing throughout to improve readability. Initial review called for a more detailed explanation of how the 3-talk framework was applied to develop the tool. The 3 steps of this tool are clearly defined, but how the framework has been adapted is still not identified. Please include a sentence explaining what the 3-talk framework is in its original form and how it was subsequently mapped onto this tool. The statement added in the background to outline the current state of SDM in breast cancer screening in Spain is contradictory. It states that SDM is offered to all women undergoing screening mammography in the form of PtDAs and that PtDAs aren't routinely used in clinical appointments. Is this a guideline vs. implementation divide or is this meant to indicate that patients use PtDAs autonomously but aren't engaged in SDM with a provider in the clinical setting? Please clarify this statement. "In Spain there is no evidence that explicitly recommends in which type of women SDM should be performed, however its use through PtDAs has been extended to all women who must decide to have mammography because they are integrated into breast cancer screening programmes in each Autonomous Community. However, PtDAs are not commonly used in clinical appointments (5)." Ensure that the study objective is consistent across the paper. In the abstract and background, the purpose is stated as to: "determine the usefulness and relevance of the contents and design provided in a handbook...". At the beginning of the methods section the purpose implied is to engage experts in the development of the tool: "it was decided to use it since when you want developing training competencies, tools to support clinical practice or a response to a
------------------	---

	professional issue, seeking the opinion of experts is a common approach (16) in this case experts are required for the development of a manual and guide”. Please indicate if examining the usefulness and relevance is to facilitate development of the tool in the background and abstract, or match the methods statement to the stated objective. There is a statement in the methods section that claims there are no SDM resources (manual and guide) in breast cancer screening. SDM in breast cancer screening is a robust field, indicate here the context in which there is a paucity of resources. “because there are no materials of this nature on the subject of SDM in breast cancer screening, and therefore it is an area without previous references.”
--	---

VERSION 2 – AUTHOR RESPONSE

Reviewer 2: Dr. Yvonne Bombard, Li Ka Shing Knowledge Institute, St. Michaels' Hospital, University of Toronto Institute of Health Policy Management and Evaluation

COMMENTS TO THE AUTHORS	AUTHORS ANSWERS
Thank you for considering the reviewer feedback. We appreciate the thoughtfulness taken to add clarifications and revisions. There remain some points of concern with the study that haven't been sufficiently addressed. In addition, the manuscript needs careful copy-editing throughout to improve readability.	Thank you very much for your comments, which we have addressed in their entirety to respond to your concerns. We hope that with the changes made they will achieve the standards of a high-quality scientific publication. We send the article to language reviewer.
Initial review called for a more detailed explanation of how the 3-talk framework was applied to develop the tool. The 3 steps of this tool are clearly defined, but how the framework has been adapted is still not identified. Please include a sentence explaining what the 3-talk framework is in its original form and how it was subsequently mapped onto this tool.	We have given a brief description in the introduction to The Tree-talk model: its 3 steps and the context for which it was created (generic health context), versus the application we give it for a specific breast cancer context and the three steps (name change in step 2) The model was created so that three key steps (1- Team Talk, 2- Option Talk, 3- Decision Talk) would be quickly grasped and to explain in an easy way how to apply SDM in generic health context for healthcare professionals (17). In this article we are adapting the three steps of Model to specific health context in BC screening to: 1) Team talk; 2) Option talk and exploring preferences; 3) Decision talk. page 3
The statement added in the background to outline the current state of SDM in breast cancer screening in Spain is contradictory. It states that SDM is offered to all women undergoing screening mammography in the form of PtDAs and that PtDAs aren't routinely used in clinical appointments. Is this a guideline vs. implementation divide or is this meant to indicate that patients use PtDAs autonomously but aren't	There was certainly a contradiction in the wording. The PtDAs was bee created for Spain context, but not general using. Also, we included two new references -“Boletín Oficial del Estado. Ley 14/1986 de 2 de febrero, General de Sanidad. BOE de 21/2000 [Official State Gazette. Law 14/1986 of 2 February, General Health. BOE of 21/2000]. Available from: https://www.boe.es/boe/dias/2001/02/02/pdfs/A04121-04125.pdf” (5)

engaged in SDM with a provider in the clinical setting? Please clarify this statement. “In Spain there is no evidence that explicitly recommends in which type of women SDM should be performed, however its use through PtDAs has been extended to all women who must decide to have mammography because they are integrated into breast cancer screening programmes in each Autonomous Community. However, PtDAs are not commonly used in clinical appointments (5).”	- “Decision aids for breast cancer screening in women approximately 50 years of age: A systematic review and meta-analysis of randomized controlled trials” 2021 (6). The phrase was edited: In Spain, Law 21/2000 Health Information Rights, Patient Autonomy and Clinical Documentation (5) protects the right to decide freely. However, SDM is not explicitly recommended for screening programs. And the scientific community are making efforts to create PtDAs (6,7) to be integrated in the Early Detection Programmes of Autonomous Communities, but, at the moment, its use is not widespread.” (page 3)
Ensure that the study objective is consistent across the paper. In the abstract and background, the purpose is stated as to: “determine the usefulness and relevance of the contents and design provided in a handbook...”. At the beginning of the methods section the purpose implied is to engage experts in the development of the tool: “it was decided to use it since when you want developing training competencies, tools to support clinical practice or a response to a professional issue, seeking the opinion of experts is a common approach (16) in this case experts are required for the development of a manual and guide”. Please indicate if examining the usefulness and relevance is to facilitate development of the tool in the background and abstract or match the methods statement to the stated objective.	The objective stated in methods (page 4) “The Delphi technique has the main objective of reaching consensus among experts on specific topics.” corresponds to aim what the literature mentions for the Delphi technique. Which helps us meet the goal of our study and then it is an explanation of why it was decided to use this technique to meet the general objective of the study (thus responding to the reviewers' comments). So, we change the redaction of the general objective for relate it better to the goal of the Delphi technique (page 2 y 4): “A Delphi method will be used to reach an agreement among experts on the contents and design of a manual and guide, designed by the research team, and to be used by health professionals in the application of SDM in breast cancer screening”
There is a statement in the methods section that claims there are no SDM resources (manual and guide) in breast cancer screening. SDM in breast cancer screening is a robust field, indicate here the context in which there is a paucity of resources. “because there are no materials of this nature on the subject of SDM in breast cancer screening, and therefore it is an area without previous references.”	The above-mentioned statement was changed to a less categorical one in the absence of documents, and was edited as follows: “in this case experts are required for the development of a manual and guide because there are few documents focused on health professionals explaining the application of SDM, specifically for breast cancer screening.” (page 4)

VERSION 3 – REVIEW

REVIEWER	Bombard, Yvonne Li Ka Shing Knowledge Institute, St. Michaels' Hospital
--

REVIEW RETURNED	04-Jan-2022
GENERAL COMMENTS	Thank you for including the suggested revisions. The manuscript needs careful copy-editing throughout to improve readability, but is otherwise acceptable for publication.